# YOLOv5-SA-FC: A Novel Pig Detection and Counting Method Based on Shuffle Attention and Focal Complete Intersection over Union

**DOI:** 10.3390/ani13203201

**Published:** 2023-10-13

**Authors:** Wangli Hao, Li Zhang, Meng Han, Kai Zhang, Fuzhong Li, Guoqiang Yang, Zhenyu Liu

**Affiliations:** School of Software, Shanxi Agricultural University, Jingzhong 030801, China; haowangli@sxau.edu.cn (W.H.); z20213621@stu.sxau.edu.cn (L.Z.); hanm@hdu.edu.cn (M.H.); sxauwangzhang@stu.sxau.edu.cn (K.Z.); lifuzhong@sxau.edu.cn (F.L.); lzysyb@sxau.edu.cn (Z.L.)

**Keywords:** pig, detection, counting, shuffle attention, focal loss

## Abstract

**Simple Summary:**

We propose a new model, YOLOv5-SA-FC, for efficient pig population detection and counting in intelligent breeding. Traditional manual methods are slow and inaccurate. Our model incorporates shuffle attention (SA) and Focal-CIoU (FC) for an improved performance. SA enhances feature extraction without adding parameters, and FC reduces the sample imbalance impact. Our experiments show that YOLOv5-SA-FC achieves a 93.8% mean average precision (mAP) and 95.6% count accuracy, outperforming other methods by 10.2% and 15.8% in pig detection and counting. This validates its effectiveness in intelligent pig breeding.

**Abstract:**

The efficient detection and counting of pig populations is critical for the promotion of intelligent breeding. Traditional methods for pig detection and counting mainly rely on manual labor, which is either time-consuming and inefficient or lacks sufficient detection accuracy. To address these issues, a novel model for pig detection and counting based on YOLOv5 enhanced with shuffle attention (SA) and Focal-CIoU (FC) is proposed in this paper, which we call YOLOv5-SA-FC. The SA attention module in this model enables multi-channel information fusion with almost no additional parameters, enhancing the richness and robustness of feature extraction. Furthermore, the Focal-CIoU localization loss helps to reduce the impact of sample imbalance on the detection results, improving the overall performance of the model. From the experimental results, the proposed YOLOv5-SA-FC model achieved a mean average precision (mAP) and count accuracy of 93.8% and 95.6%, outperforming other methods in terms of pig detection and counting by 10.2% and 15.8%, respectively. These findings verify the effectiveness of the proposed YOLOv5-SA-FC model for pig population detection and counting in the context of intelligent pig breeding.

## 1. Introduction

With the advancement of agricultural informatization, the pig farming industry is undergoing a rapid transformation towards intensification, scale, and intelligence. The dynamic nature of pig farming necessitates accurate and efficient pig detection and counting methods.

However, the efficient and accurate detection and counting of pigs still pose significant challenges to the present day. There are several reasons for this. First, as the farming industry continues to expand, the number of pigs in pens has been increasing. Second, pigs can become dirty due to their various behaviors, and their tendency to cluster and nest can result in a large amount of occlusion, indistinct body features, and difficulty in distinguishing them from the environment [1]. Additionally, changes in pig numbers occur due to factors such as deaths, sales, new pigs entering the herd, pen splitting or merging, and pigs growing into the next stage [2]. Therefore, there is an urgent requirement for a pig counting method that can maintain a certain level of accuracy and efficiency in high-density dynamic farming environments, in order to truly achieve intensification and intelligence of the industry.

At present, pig counting in the industry relies heavily on manual inspections, which are known to be time-consuming, labor-intensive, and inefficient [3]. Moreover, factors such as pigs moving back and forth can significantly reduce the accuracy of manual counting. Additionally, increased contact between caretakers and pigs during manual counting increases the risk of transmission of zoonotic diseases.

While electronic ear tags have been utilized for counting [4,5], they come with their own set of challenges. Pigs coming into contact with each other can lead to false reports, and there is a risk of ear tags falling off or getting damaged in environments where pigs scratch or in muddy conditions.

In the field of computer vision, Tian et al. [6] have proposed an innovative method for pig counting on farms using deep learning techniques. Their approach combines a counting CNN and the ResNeXt model, achieving a high level of accuracy while maintaining a low computational cost. The results demonstrated a mean absolute error of 1.67 when applied to real-world data. Jensen et al. [7] developed a novel approach for the automatic counting and positioning of slaughter pigs within a pen, utilizing a convolutional neural network (CNN) with a single linear regression output node. This model receives three partial images corresponding to different areas of the pen and estimates the number of pigs in each area. Furthermore, they obtained promising results, with a mean absolute error of less than one pig and a coefficient of determination between estimated and observed counts exceeding 0.9.

To enhance the automated piglet counting performance and address the challenge of partial occlusion, Huang et al. [8] have proposed a two-stage center clustering network (CClusnet). In the initial stage, CClusnet predicts a semantic segmentation map and a center offset vector map for each image. In the subsequent stage, these maps are combined to generate scattered center points, and the piglet count is obtained using the mean-shift algorithm. This method achieved a mean absolute error of 0.43 per image for piglet counting. A bottom–up pig counting algorithm detected and associated three kinds of keypoints to count pigs [9]; however, the use of this method can be challenging due to the possibility of occlusion and keypoints being invisible. The use of farrowing stalls exacerbates the difficulty of counting, as occlusion can cause a piglet to appear to be fragmented into multiple smaller parts within the scene, making it even more challenging to accurately count piglets.

Building upon traditional computer vision technology, Kashila et al. [10] have utilized elliptical displacement calculation methods to achieve an impressive accuracy of 89.8% in detecting pig movement. In another study by Kashila et al. [11], they employed an ellipse fitting technique to obtain a high accuracy of 88.7% in detecting and identifying individual pigs. Nasirahmadi et al. [12,13] have also utilized ellipse fitting and the Otus algorithm to successfully detect individual pigs and accurately determine their lying positions.

Tu et al. [14] have proposed an innovative pig detection approach for grayscale video utilizing foreground object segmentation. Their method involves three stages. First, texture information is integrated to construct the background model. Next, pseudo-wavelet coefficients are computed, which are utilized in the final stage to estimate a probability map using a factor graph and a loopy belief propagation (BP) algorithm. However, it is important to note that this method suffers from high computational complexity due to the use of the BP algorithm and factor graphs. In a different study [15], a background subtraction method based on a Gaussian Mixture Model (GMM) [16] was utilized to detect moving pigs in scenarios with no windows and continuous lighting for 24 h. It is worth mentioning that the GMM background subtraction method can be computationally intensive and time-consuming. To address the limitations of the GMM approach, Li et al. [17] have developed an enhanced pig detection algorithm based on an adaptive GMM. In this method, the Gaussian distribution is scanned periodically—typically once every m frames—in order to adapt the model. Redundant Gaussian distributions are detected and eliminated to enhance the convergence speed of the model. However, it is important to note that this method may face challenges in detecting pigs when sudden lighting changes occur. Traditional computer vision technology has the potential to improve animal welfare and achieve high recognition accuracy; however, it may not be suitable for industrial production requirements due to its slow detection speed. Additionally, the performance of the model may notably drop when the pigs are occluded or when there is significant variation in the size of the target pigs in the image.

To further enhance accuracy, numerous researchers have leveraged deep neural networks for pig detection and counting. Marsot et al. [18] have employed a two-step approach for pig recognition. They first utilized two Haar feature-based cascade classifiers and a shallow convolutional neural network to automatically detect pig faces and eyes. Subsequently, a deep convolutional neural network was employed for recognition. Their approach achieved an accuracy of 83% on a test set consisting of 320 images containing 10 pigs.

Martin et al. [19] have employed the Faster R-CNN [20] object detection pipeline and the Neural Architecture Search (NAS) backbone network for feature extraction, achieving a mean average precision (mAP) of 80.2%. In another study conducted by the same authors [21], Faster R-CNN was utilized to detect the positions and poses of pigs in all-weather videos captured from 10 pigsties. The detection performance yielded an mAP of 84% during the day and 58% during the night. Zhang et al. [22] employed three CNN-based models—namely, the Faster-RCNN [20], R-FCN [23], and SSD models [24]—for individual pig detection. Similarly, van der Zande et al. [25] have utilized the YOLOv3 model for the same purpose. In another study, conducted by Guo et al. [26], the YOLOv5s model was employed to achieve automated and continuous individual pig detection and tracking.

Although deep learning methods have achieved promising results in terms of pig detection, the use of attention mechanisms for feature extraction has not yet been fully explored. Additionally, the used loss functions may not effectively constrain the detection process to ensure precise results. To address these challenges and improve the pig detection and counting performance, we focused on exploring breakthroughs in the following areas.

Shuffle attention [27] is an attention mechanism that integrates group convolutions, spatial attention mechanisms, channel attention mechanisms, and the concepts of ShuffleNet. By introducing channel shuffle operations and block-wise parallel usage of spatial and channel attention mechanisms, shuffle attention achieves an efficient and tight integration of the two attention mechanisms while also possessing the characteristics of a low computational cost and plug-and-play capability. This means that shuffle attention can be quickly and seamlessly integrated into any CNN architecture for training while ignoring computational cost overheads.

Focal-CIoU is an advanced variant of CIoU, which aims to resolve the issue that CIoU may fail to accurately reflect the true differences in an object’s width, height, and confidence level. By introducing a focal term into the CIoU loss function, Focal-CIoU effectively improves upon the performance of CIoU and achieves more accurate results in object detection.

Leveraging the advantages of the various methods mentioned above, we propose a network model that integrates the shuffle attention mechanism and Focal-CIoU into YOLOv5, with the aim of achieving effective detection and counting of densely raised pigs. Specifically, in contrast to density map-based counting methods, YOLOv5-based counting directly detects the size and location of each target, allowing for accurate counting of pigs based on the identified targets. By utilizing YOLOv5-based counting, it becomes possible to directly annotate and visualize the pigs in the original image. This approach enhances our understanding of their behavior and simplifies the detection of movement patterns. To verify the effectiveness of the proposed model, several comparative experiments were designed to compare the performance differences between different models and YOLOv5-SA-FC. The main innovations of this paper can be summarized as follows:We first establish a novel data set for pig detection and counting, which comprises 8070 images. The original videos were captured from six cameras installed on a farm over a period of two months. There are more than 200 pigs on the farm, with ages ranging from around 140 to 150 days. The aforementioned factors provide a more diverse data set, including variations in illumination, ages, angles, and other aspects.We propose a novel pig detection and counting method called YOLOv5-SA-FC, which is based on the shuffle attention module and the Focal-CIoU loss function. The channel attention and spatial attention units in the shuffle attention module enable YOLOv5-SA-FC to focus on regions in the image that are crucial for detection, leading to the extraction of more rich, robust, and discriminative features. Meanwhile, the Focal-CIoU loss function ensures that the proposed YOLOv5-SA-FC model emphasizes prediction boxes having higher overlap with the target box, leading to an increased contribution of positive samples and improved model performance. Our model achieves the best performance for pig detection and counting tasks, with a 2.3% improvement over existing models.We conducted several comparative and ablation experiments to validate the performance of our proposed model, including a comparison with different models, evaluations of the effectiveness of the shuffle attention module and the Focal-CIoU loss function, and an overall assessment of the superiority of YOLOv5-SA-FC.

The remainder of this paper is organized as follows. In Section 2, we provide a detailed description of the materials and methods used in this study. In Section 3, we present the results and discussions based on the conducted experiments. Section 4 discusses the implications of the results obtained in our study. Finally, we conclude our findings and summarize the contributions of this paper in Section 5.

## 2. Materials and Methods

### 2.1. Data Set

The data utilized in this experiment were gathered from Nonglueyuan Farm, located in Xiangfen County, Linfen City, Shanxi Province, China. The farm has an enclosed environment formed by railings that create an enclosed circular house, the ground of which is made of concrete, and some portions are constructed in a striped pattern. A fixed camera from the Hikvision DS-2DE3Q120MY was used to capture images of the pigs. The camera was installed at a height of approximately 170 cm above the ground and was pointed towards the inside of the pigsty. The data collection phase lasted for two months, from August to October 2022. Please note that videos with poor picture quality due to factors such as lighting conditions were excluded. Overall, a total of approximately 2 Terabytes of video data were obtained.

To obtain an effective model for pig detection and counting, we processed the collected pig videos as follows. First, we selected videos with clear images and captured one image every 20 s, saving them in JPG format. Second, in order to ensure the validity of the collected images and facilitate model training and validation, we manually screened all of the captured images and removed blurry and highly repetitive images. Third, we used the Make Sense.ai annotation tool for online annotation and saved the annotated data as TXT files. Some sample images are shown in Figure 1. After processing, we obtained a data set containing 8070 images, with an average of about 15 target pigs per image. Some sample images are shown in Figure 2. Figure 2a indicates an image taken with the camera positioned above the farm at a 45° angle, while Figure 2b shows another with the camera positioned diagonally at a 45° angle. To evaluate the performance of the proposed model, the data set was divided as follows: 6955 samples were used for training and 1115 samples were used for testing. Additionally, to increase the diversity of the data and allow the model to capture richer features, we employed various data augmentation techniques, including mosaic, random horizontal flipping, scaling, HSV color space transformation, and translation. Some sample images are shown in Figure 3.

HSV (Hue, Saturation, Value) [28] is a color space widely used in image processing and data augmentation, particularly in computer vision and deep learning tasks, such as image classification, object detection, and semantic segmentation. It plays a significant role in these fields. HSV techniques have various applications in data augmentation. Firstly, through color jittering, the values of the HSV channels can be randomly adjusted, creating new images slightly different from the original, thereby increasing the diversity of the data. This method effectively enriches the training data and helps improve the model’s robustness. Secondly, brightness and contrast enhancement is another way to apply HSV techniques. By adjusting the values of the brightness and saturation channels, the brightness and color saturation of the image can be increased or decreased, generating images with different lighting conditions. This method helps the model adapt to different environments. Finally, HSV transformations can also be part of data augmentation strategies. During the data augmentation process, HSV random transformations, such as random translation, rotation, and scaling, can be applied to generate more training samples. This is particularly helpful for training robust models. In summary, HSV techniques provide powerful tools for data augmentation, increasing data diversity, and improving model performances.

### 2.2. Technical Route

The proposed YOLOv5-SA-FC model follows the technical framework illustrated in Figure 4. First, the collected data are pre-processed by removing blurry images and resizing the input images to 320 × 320 px. Second, different data augmentation techniques, such as translation, scaling, mosaic, and flipping, are employed to expand the data set and increase its diversity, resulting in an improved model performance. Finally, after the pre-processing stage, the images are fed into the YOLOv5-SA-FC model for training and the detection and counting of pigs, resulting in accurate and reliable results.

### 2.3. YOLOv5-SA-FC

#### 2.3.1. YOLOv5

YOLOv5 [29] is a popular object detection algorithm that has been widely used for various tasks. Based on the network depth, YOLOv5 is available in different versions, such as YOLOv5s, YOLOv5m, YOLOv5l, and YOLOv5x. It is important to note that we chose the lightweight YOLOv5s as the baseline model for experimental validation in this paper. Specifically, YOLOv5 divides the input image into multiple grid cells, with three anchor boxes predicted for each grid cell. Each anchor box contains parameters for the height, width, anchor point coordinates, and confidence. The confidence score represents the probability of an object being present in the grid cell, which is calculated as follows:(1)$$Confidence=p_{r}\left(obj\right)\times IOU,$$
where $p_{r}\left(obj\right)\in \left[0,1\right]$ represents the probability of an object’s presence in the grid cell. *IoU* represents the Intersection over Union between the predicted bounding box and the ground-truth anchor box. The confidence score reflects the probability of the presence of an object in the grid cell and the accuracy of object detection in the prediction box when there is an object in the grid cell. Finally, non-maximal suppression (NMS) is applied to remove redundant anchor boxes, and the position and size of the corresponding anchor boxes are adjusted to generate the final model predictions.

#### 2.3.2. Shuffle Attention

The shuffle attention [27] structure is depicted in detail in Figure 5. The input feature map is first divided into groups, and for each group a shuffle unit is employed to combine the channel attention and spatial attention into a single block. Subsequently, all sub-features are aggregated, and an operator called “channel shuffle” is applied to facilitate information exchanges among different sub-features.

The grouping of features:In the SA module, given a feature map *F* with dimensions $R^{C\times H\times W}$, where *C* represents the number of channels and *H* and *W* denote the spatial height and width, respectively, the feature map is divided into *G* sub-features: $F=[F_{1};F_{2};\ldots ;F_{G}]$, where each sub-feature $F_{k}\in R^{(C/G)\times H\times W}$ captures a specific semantic response during the training process. Next, an attention module is applied to generate importance coefficients for each sub-feature. At the beginning of each attention unit, the input $F_{k}$ is split into two branches along the channels $F_{k1}$ and $F_{k2}$, both with dimensions $R^{(C/2G)\times H\times W}$, as shown in Figure 5.The channel attention branch:The channel attention branch focuses on the informative parts in terms of what they represent, rather than their specific location. Specifically, in the channel attention branch, global information is embedded by applying global average pooling (GAP) to $F_{k1}$.
(2)$$c=F_{gp}=\frac{1}{H\times W}\sum_{i=1}^{H}\sum_{j=1}^{W}F_{k1}\left(i,j\right).$$Moreover, a compact feature is generated to offer guidance for adaptive and precise selection. This is achieved using a straightforward gating mechanism that utilizes sigmoid activation. Through the application of this gating mechanism, the final output of the channel attention can be obtained, facilitating effective and accurate selection. This process can be formulated as follows:
(3)$$F_{k1}^{\prime }=\sigma \left(F_{c}\left(s\right)\right)\cdot F_{k1}=\sigma \left(W_{1}c+b_{1}\right)\cdot F_{k1},$$
where the parameters $W_{1}\in R^{(C/2G)\times 1\times 1}$ and $b_{1}\in R^{(C/2G)\times 1\times 1}$ are employed to shift and scale the channel-wise statistics *c*, respectively. These parameters allow for the adjustment of the values of *s*, enabling fine-tuning and controlling the influence of the channel attention on the final output.The spatial attention branch:Unlike channel attention, spatial attention emphasizes the informative parts in terms of where they are located, thus complementing the channel attention. The process begins by applying the Group Norm (GN) [30] to $F_{k2}$, resulting in spatial statistics. Then, Fc(·) is utilized to enhance the representation of $F_{k2}$. The final output of the spatial attention is obtained by performing the following operation:
(4)$$F_{k2}^{\prime }=\sigma \left(W_{2}\cdot GN(F_{k2})+b_{2}\right)\cdot F_{k2}.$$The two branches—that is, channel attention and spatial attention—are concatenated to ensure that the number of channels matches the number of inputs (i.e., $F_{k}^{\prime }=\left[F_{k1}^{\prime },F_{k2}^{\prime }\right]\in R^{(C/G)\times H\times W}$).Aggregation:Following the aggregation of all sub-features, a channel shuffle operator is employed to facilitate the flow of information across different groups along the channel dimension. This operator, similar to the one used in ShuffleNet v2 [27], allows for effective communication and the exchange of information between different sub-features, enhancing the overall performance of the model.

#### 2.3.3. The Proposed Novel YOLOv5-SA-FC Model

The proposed YOLOv5-SA-FC model is constructed by integrating the shuffle attention mechanism and Focal-CIoU (FC) loss into the YOLOv5 backbone network. This design improves the robustness of the model and enables the network to learn richer features. Figure 6 presents the architecture of the novel YOLOv5-SA-FC model.

BackboneThe main purpose of the backbone network is to extract features and progressively down-sample the feature maps.1.Conv: The Conv module consists of a Conv2d layer, a BatchNorm2d layer, and the Sigmoid Linear Unit (SiLU) activation function. The Conv2d layer performs the convolutional operation, which applies a set of learnable filters to the input feature map, extracting local patterns and features. The BatchNorm2d layer normalizes the output of the Conv2d layer, ensuring stable and consistent feature representations during training. The SiLU activation function is applied element-wise to the output of the BatchNorm2d layer.It is defined as follows:
(5)SiLU(x)=x∗sigmoid(x).These components of the Conv module work synergistically to extract and process features in the YOLOv5-SA-FC model.2.C3: The C3 module is also known as the Cross-Stage Partial Network (CSPNet) module with 3 convolutions. After the feature map enters the C3 module, it is split into two paths: the left path includes a Conv module and a Bottleneck module, while the right path only passes through a Conv module. Finally, the outputs of both paths are concatenated and passed through another Conv module. In the C3 module, all three Conv modules consist of 1 × 1 convolutions and serve the purpose of dimensionality reduction or expansion.3.Shuffle attention: The shuffle attention module operates by grouping the channel dimensions of the input feature map into multiple sub-features. For each sub-feature, a shuffle unit integrating two complementary attention mechanisms—channel attention and spatial attention—is employed.4.SPPF: The SPPF module is used for spatial pyramid pooling and feature fusion. It divides the input feature map into grids of different sizes and performs pooling operations to capture multi-scale information. This allows the model to gain a better understanding of objects at different scales.NeckThe neck part combines the Feature Pyramid Network (FPN) [31] structure and the Path Aggregation Network (PAN) [32] structure. From Figure 6, it can be observed that the left branch of the neck module performs up-sampling by interpolating the feature maps, increasing their scale to facilitate the fusion of features obtained from the backbone. In contrast, the right branch of the neck module continues down-sampling. This serves two purposes: to obtain feature maps at different scales and to achieve better fusion between shallow visual features and deep semantic features, going beyond simple concatenation.HeadThe head layer serves as the detection module, which has a relatively simple network structure consisting of three 1 × 1 convolutions corresponding to the three detection feature layers.

#### 2.3.4. The Loss Function

Designing an appropriate loss function is essential for optimizing a pig detection model. The considered loss function consists of three terms: locality loss ($L_{loc}$), category loss ($L_{cls}$), and confidence loss ($L_{conf}$). In the following, we will provide a detailed description of them. The loss function of YOLOv5-SA-FC is defined as follows.
(6)$$Loss=L_{loc}+L_{cls}+L_{conf}.$$

The locality $L_{loc}$ loss—Focal-CIoU loss (FC loss):In practical object detection scenarios, there is often a severe class imbalance between positive samples (i.e., bounding boxes containing objects) and negative samples (i.e., bounding boxes not containing objects). The default locality loss (i.e., CIoU loss) treats all samples equally, which fails to effectively address this issue. Consequently, the model will tend to overly focus on prediction boxes having less overlap with the ground truth, resulting in a degradation of the model performance. This is primarily due to the dominance of negative samples in the weight contribution during the optimization process, whereas accurate prediction of positive samples is desired. To tackle this problem, a novel locality loss based on the CIoU loss function, called Focal-CIoU, is introduced. By resetting the weights in L_CIoU based on the IoU values, Focal-CIoU increases the contribution of positive samples in L_CIoU:
(7)$$L_{loc}=L_{Focal-CIoU}=IoU^{\gamma }\cdot L_{CIoU},$$
where IoU denotes the Intersection over Union, $L_{CIoU}$ indicates the CIoU loss [33], and the parameter γ (as mentioned in [33]) controls the curvature of the curve and determines the degree of outlier suppression. The default value for γ is 0.5. Among these, IoU and $L_{CIoU}$ are defined as follows:
(8)$$IoU=\frac{\left|A\cap B\right|}{\left|A\cup B\right|},$$IoU measures the overlap between the predicted bounding box (denoted by *A*) and the ground-truth bounding box (denoted by *B*), quantifying the extent to which the predicted region aligns with the ground-truth region.
(9)$$L_{CIoU}=1-IoU+\frac{\rho^{2}\left(b,b^{gt}\right)}{c^{2}}+\alpha v,$$
where *b* and $b^{gt}$ denote the center points of the predicted and ground-truth bounding boxes, respectively; ρ represents the Euclidean distance between the two center points; *c* represents the diagonal length of the minimum enclosing rectangle that contains both the predicted and ground-truth bounding boxes; *v* is employed to quantify the consistency of aspect ratios [34]; and α is a weight function. The definitions of *v* and α are as follows:
(10)$$v=\frac{4}{\pi^{2}}\left(arctan\frac{w^{gt}}{h^{gt}}-arctan\frac{w}{h}\right),$$
(11)$$\alpha =\frac{v}{1-IoU}+v,$$
where $w^{gt}$ and $h^{gt}$ represent the width and height of the ground-truth bounding box, while *w* and *h* represent the width and height of the predicted bounding box, respectively.The category loss $L_{cls}$:The category loss is calculated based on the cross-entropy loss function and can be obtained as follows:
(12)$$L_{cls}=-\sum_{i=0}^{s^{2}}\sum_{j=0}^{B}I_{i,j}^{obj}\sum_{c\in cls}\left[{\bar{P}}_{i}^{j}\left(c\right)log\left(P_{i}^{j}\left(c\right)\right)+\left(1-{\bar{P}}_{i}^{j}\left(c\right)\right)log\left(1-P_{i}^{j}\left(c\right)\right)\right],$$
where $s^{2}$ indicates an s×s grid; *B* represents the number of bounding boxes predicted in each grid; $I_{i,j}^{obj}$ is an indicator that is equal to 1 when a target is present in the *j*th box and 0 otherwise; and ${\bar{P}}_{i}^{j}\left(c\right)$ and $P_{i}^{j}\left(c\right)$ represent the predicted and ground-truth probabilities, respectively, of an object in the *j*th box of the *i*th grid belonging to the *c*th class.The confidence loss $L_{conf}$:The confidence loss is obtained through the following equations:
(13)$$\begin{array}{c} L_{conf}=-\sum_{i=0}^{s^{2}}\sum_{j=0}^{B}I_{i,j}^{obj}\left[{\bar{V}}_{i}^{j}log\left(V_{i}^{j}\right)+\left(1-{\bar{V}}_{i}^{j}\right)log\left(1-V_{i}^{j}\right)\right]\\ -\lambda_{noobj}\sum_{i=0}^{s^{2}}\sum_{j=0}^{B}I_{i,j}^{noobj}\left[{\bar{V}}_{i}^{j}log\left(V_{i}^{j}\right)+\left(1-{\bar{V}}_{i}^{j}\right)log\left(1-V_{i}^{j}\right)\right], \end{array}$$
where $s^{2}$, *B*, and $I_{i,j}^{obj}$ have similar meanings as in Equation (Equation 12); $I_{i,j}^{noobj}$ is an indicator that is equal to 0 when a target is present in the *j*th box and 1 otherwise; $\bar{V}$ and *V* denote the confidence values for the predicted and annotated boxes, respectively; and $\lambda_{noobj}$ is a hyperparameter that is utilized to balance the importance of the two terms.

#### 2.3.5. Evaluation Metric

In order to effectively evaluate the performance of the proposed model, we utilize several evaluation metrics in this paper, including precision (Equation (Equation 14)), recall (Equation (Equation 15)), F1 score (Equation (Equation 16)), average precision (Equation (Equation 17)), and mean average precision (mAP; Equation (Equation 18)). These metrics are defined as follows:(14)P=TP/(TP+FP),
(15)R=TP/(TP+FN),

In Equations (Equation 14) and (Equation 15), TP represents true positive, which indicates that there is a pig in the image and the algorithm correctly predicts its presence; FP stands for false positive, indicating that there is no pig in the image, but the algorithm incorrectly detects one; and FN represents false negative, indicating that the algorithm fails to detect a pig that is actually present in the image. To determine whether an object is considered a true positive, the algorithm calculates the Intersection over Union (IoU) between the predicted bounding box and the ground-truth bounding box. If the IoU is greater than a specified threshold (e.g., IoU > 0.5), the object is considered a true positive. Objects with an IoU below the threshold are considered false positives, while those that are not correctly identified are considered false negatives.

Precision (P), calculated using Equation (Equation 14), measures the proportion of correctly predicted positive instances out of all instances predicted as positive. It provides an indication of the model’s accuracy in identifying true positives. Recall (R), which is calculated using Equation (Equation 15), measures the proportion of correctly predicted positive instances out of all actual positive instances, thus indicating the model’s ability to capture all positive instances.
(16)$$F1\_score=\frac{2PR}{P+R},$$

The F1_score, determined using Equation (Equation 16), is the harmonic mean of precision and recall, providing a balanced measure of the model’s performance by considering both precision and recall, where P and R are from Equations (Equation 14) and (Equation 15).
(17)$$AP=\int_{0}^{1}P(R)dR,$$

Average precision (AP), computed using Equation (Equation 17), is the average of precision values at different recall levels, which provides a comprehensive measure of the model’s performance across various recall thresholds, where $\int_{0}^{1}$ indicates the integration over data points on the precision–recall curve within the range from 0 and 1. P(R) represents precision at each recall point (r). In other words, P(R) is the precision of the model at a specific recall level.
(18)$$mAP=\frac{\sum_{1}^{n}\left(AP\right)}{n},$$

Finally, the mean average precision (mAP), calculated using Equation (Equation 18), is the average of average precision values across different classes or categories, which provides an overall assessment of the model’s performance in object detection tasks, where *n* signifies the total number of categories. This fraction is used to normalize the AP values for each category, ensuring that different categories contribute equally to the mAP. These evaluation metrics collectively assess the effectiveness and accuracy of the proposed model in detecting and counting objects.

#### 2.3.6. Experimental Setup

This study was carried out using a Linux Ubuntu 18.04 operating system with the PyTorch deep learning framework and Python programming language. The hardware used for the experiments included an Intel Core I7 7800 X CPU, NVIDIA GeForce GTX TITANXP GPU, and 128 GB memory. For model training, the iteration count was set to 100, the batch size was set to 16, the initial learning rate was set to 0.01, and the learning rate momentum was set to 0.937. Further details on the hardware and software configuration used in the experiments are provided in Table 1.

The training process for the YOLOv5-SA-FC model is detailed in Algorithm 1.
**Algorithm 1:** YOLOv5-SA-FC model training
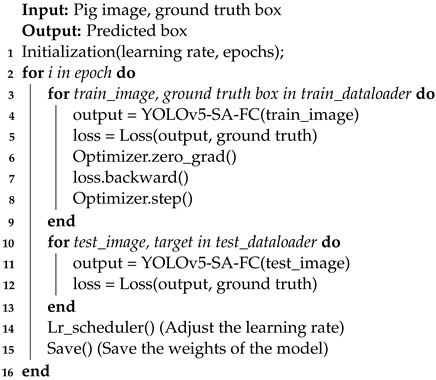


## 3. Experimental Results and Analyses

In this section, we detail our experimental results and provide relevant discussions. The experiments are divided into several parts, including a comparison of different models, evaluations of the effectiveness of the shuffle attention module and the Focal-CIoU loss module, and an overall evaluation of the superiority of the YOLOv5-SA-FC model.

### 3.1. Comparison of Different Models

To verify the effectiveness of our proposed YOLOv5-SA-FC model, we compared it with several other models, including Faster-RCNN [20], YOLOv2 [35], YOLOv3 [36], YOLOv4 [37], and YOLOv5. The results are presented in Table 2.

According to Table 2, our proposed YOLOv5-SA-FC model achieved the best performance across all evaluation criteria. Specifically, YOLOv5-SA-FC achieved a counting accuracy of 95.6%, which is 36.96%, 19.80%, 15.46%, and 10.90% higher than the YOLOv2, YOLOv3, YOLOv4, and YOLOv5 models, respectively. In addition, YOLOv5-SA-FC achieved a precision of 92.7%, which is 12.9%, 47.14%, 6.67%, 10.36%, and 1.87% better than the Faster-RCNN, YOLOv2, YOLOv3, YOLOv4, and YOLOv5 models, and a recall of 88.1%, which is 7.70%, 29.37%, 24.08%, 16.84%, 6.14%, and 4.51% better than the Faster-RCNN, SSD, YOLOv2, YOLOv3, YOLOv4, and YOLOv5 models. Moreover, YOLOv5-SA-FC achieved a 0.90 F1 score, which is 0.11, 0.13, 0.34, 0.12, 0.80, and 0.30 higher than the Faster-RCNN, SSD, YOLOv2, YOLOv3, YOLOv4, and YOLOv5 models, and an mAP of 93.8%, which is 4.80%, 8.31%, 35.55%, 12.20%, 5.87%, and 3.08% superior to those of the aforementioned models, respectively. These results clearly demonstrate the effectiveness of our proposed YOLOv5-SA-FC model.

Figure 7 illustrates the comparison of different models across various iterations. The curves representing the mAP, recall, and F1 score of YOLOv5-SA-FC consistently outperform those of the Faster-RCNN, SSD, YOLOv2, YOLOv3, YOLOv4, and YOLOv5 models, indicating that YOLOv5-SA-FC had a superior performance, in terms of these metrics, when compared to the other models. We should note that the precision of the SSD was superior to that of our model, which was due to the more complex architecture of the SSD, specifically in terms of its feature extraction network. Additionally, the SSD utilizes a series of prior boxes (anchors) for object detection, which allows it to cover objects of different scales and aspect ratios. However, these factors also contribute to the slower speed of the SSD. Taking into account both speed and performance considerations, the obtained results validate the effectiveness of YOLOv5-SA-FC throughout the entire training phase.

Furthermore, the qualitative comparison results for the different models, including the Faster-RCNN, SSD, YOLOv2, YOLOv3, YOLOv4, and YOLOv5 models, are depicted in Figure 8. It is evident that YOLOv5-SA-FC outperformed the other models by obtaining more accurate predicted bounding boxes, especially in scenarios involving occlusion and other challenging situations.

The superiority of the YOLOv5-SA-FC model over other models can be attributed to the following points. The shuffle attention and Focal-CIoU loss used in YOLOv5-SA-FC enabled it to adaptively focus on more discriminative regions of the image for pig detection, allowing it to effectively fuse feature maps at various scales and extract more informative features, even in scenarios involving pig clustering or occlusion. As a result, YOLOv5-SA-FC was shown to be capable of achieving rapid and efficient pig detection and counting, with a remarkable 95.6% counting accuracy and 93.8% mAP—significantly better than those for the other models.

### 3.2. Evaluating the Effectiveness of the Shuffle Attention Module

To assess the effectiveness of the shuffle attention module, we conducted a comparison between YOLOv5 and YOLOv5-SA. YOLOv5-SA was formed by integrating the shuffle attention module into the YOLOv5 backbone. The results are given in Table 3.

Table 3 shows that the YOLOv5-SA model outperforms the YOLOv5 model on all evaluation criteria. Specifically, YOLOv5-SA achieved a 94.4% counting accuracy, which is 9.51% higher than that of YOLOv5; 92.3% precision, which is 1.43% better than that of YOLOv5; 87.9% recall, which is 4.27% superior to that of YOLOv5; and a 0.89 F1 score, which is 2.30% better than that of YOLOv5. Additionally, YOLOv5-SA achieved an mAP of 93.6%, which is 2.86% higher than that of YOLOv5. These results demonstrate the effectiveness of the shuffle attention module.

Moreover, Figure 9 illustrates the results obtained by the different models at various iterations. The curves representing the mAP, precision, recall, and F1 score for YOLOv5-SA consistently outperformed those of YOLOv5, indicating that YOLOv5-SA had a superior performance in terms of mAP, precision, recall, and F1 score, when compared to YOLOv5. These results validate the effectiveness of the shuffle attention module throughout the entire training phase.

The superiority of the YOLOv5-SA model over the YOLOv5 model can be attributed to several reasons. First, the inclusion of spatial and channel attention mechanisms in YOLOv5-SA enabled it to focus on both the informative parts and their spatial locations, leading to more accurate predictions. This attention mechanism allows the model to dynamically adapt its attention to different regions of the input, improving its ability to capture relevant features. Second, YOLOv5-SA employs feature fusion techniques and a channel shuffle operator to facilitate the integration of information across different groups, enabling the model to capture diverse features and, thus, enhancing its performance. The channel shuffle operation promotes a cross-group information flow, allowing for better communication and exchanges of information between different parts of the network. Overall, the combination of spatial attention, channel attention, feature fusion, and the channel shuffle operator in YOLOv5-SA led to improved accuracy and efficiency, making it the more effective and advanced model in the comparison.

### 3.3. Evaluating of the Effectiveness of the Focal-CIoU Loss Function

To confirm the superiority of our proposed YOLOv5-SA-FC model, we carried out a comparison with the other models mentioned above, including YOLOv5, YOLOv5-SA, and YOLOv5-FC, against YOLOv5-SA-FC, which combines the shuffle attention and Focal-CIoU modules into the YOLOv5 structure. The results are presented in Table 4.

Table 5 clearly demonstrates that the YOLOv5-FC model outperforms the YOLOv5 model across all evaluation criteria. Specifically, YOLOv5-FC achieved an impressive counting accuracy of 88.8%, which is 3.02% higher than that of YOLOv5. Additionally, it achieved a precision of 92.2%, which is 1.32% better than that of YOLOv5, and a recall of 85.8%, which is 1.78% superior to that of YOLOv5. Moreover, YOLOv5-FC achieved an F1 score of 0.87, which is 2.29% better than that of YOLOv5. Finally, YOLOv5-SA achieved an mAP of 92.3%, which is 1.43% higher than that of YOLOv5. These results clearly demonstrate the effectiveness of the shuffle attention module.

Additionally, Figure 10 presents the comparison results of different models at various iterations. The curves depicting mAP, precision, recall, and F1 scores consistently outperform for YOLOv5-FC in comparison to YOLOv5. This clearly indicates that YOLOv5-FC exhibits a superior performance across mAP, precision, recall, and F1 score metrics when compared to YOLOv5. These results serve as validation for the effectiveness of the Focal-CIoU loss throughout the entire training phase.

The YOLOv5-FC model has been shown to outperform YOLOv5 due to several key factors. One of the main advantages of the Focal-CIoU loss function is that it addresses the issue of class imbalance while also improving the localization accuracy of the object detection model. Additionally, this loss function can effectively reduce the impact of easy negative samples, which are samples that are clearly not objects of interest. By doing so, the model can better focus on the more challenging samples that are critical for accurate object detection. Overall, the YOLOv5-FC model offers significant improvements over its predecessor, making it a powerful tool for object detection tasks.

### 3.4. Evaluating the Superiority of YOLOv5-SA-FC

To confirm the superiority of our proposed YOLOv5-SA-FC model, we conduct a comparison with other models mentioned in the previous section, such as YOLOv5, YOLOv5-SA, and YOLOv5-FC. We also introduce YOLOv5-SA-FC, which combines the shuffle attention and Focal-IoU modules into the YOLOv5 structure. The comparison results are presented in Table 4.

Table 4 displays a comparison of the performance metrics for different models, highlighting the effectiveness of our proposed YOLOv5-SA-FC model. Our model outperforms others across all evaluation criteria, as demonstrated by Table 4. Specifically, YOLOv5-SA-FC showcases significant advantages over YOLOv5, YOLOv5-FC, and YOLOv5-SA models. In terms of counting accuracy, it achieves an impressive 95.6%, surpassing YOLOv5 by 10.90%, YOLOv5-FC by 7.66%, and YOLOv5-SA by 1.27%. Additionally, in precision our model achieves 92.7%, outperforming YOLOv5 by 1.87%, YOLOv5-FC by 0.54%, and YOLOv5-SA by 0.43%. Moreover, in recall YOLOv5-SA-FC achieves 88.1%, which is 4.51%, 2.68%, and 0.23% higher than YOLOv5, YOLOv5-FC, and YOLOv5-SA, respectively. Further emphasizing its superiority, YOLOv5-SA-FC attains a 0.90 F1 score, showcasing improvements of 3.44%, 1.12%, and 1.12% over YOLOv5, YOLOv5-FC, and YOLOv5-SA, respectively. Additionally, its mean average precision (mAP) of 93.8% surpasses the corresponding values for the aforementioned models by 3.08%, 1.62%, and 0.21%.

Furthermore, Figure 11 showcases the comparison results of different models at various iterations. The curves representing mAP, precision, recall, and F1 scores show YOLOv5-SA-FC consistently outperforms those of other models. This clearly indicates that YOLOv5-SA-FC exhibits a superior performance across all metrics when compared to the other models. These results serve as strong validation for the effectiveness of YOLOv5-SA-FC throughout the entire training phase.

A heatmap comparison of the different models is provided in Figure 12. The heatmaps illustrate the differences in the activation patterns and highlight the areas of focus for each model. The heatmap of YOLOv5-SA-FC presents more precise and concentrated activations, indicating its ability to accurately identify and localize objects. In contrast, the heatmaps of other models show scattered, less distinct activations. Additionally, the other models were prone to missing detections in the case of occlusion, as can be observed from the YOLOv5 heatmap. This comparison intuitively demonstrates the superior performance of YOLOv5-SA-FC in terms of capturing relevant features and making accurate predictions.

The YOLOv5-SA-FC model was found to outperform the other models for several reasons. The shuffle attention mechanism allows the model to selectively focus on informative features while suppressing irrelevant ones, thus reducing the impact of noisy or irrelevant information and improving the robustness of the model. The Focal-CIoU loss function addresses the issue of class imbalance in object detection, which is common in real-world scenarios as certain classes tend to be rare or under-represented. It assigns higher weights to hard examples and reduces the influence of easy ones, improving the model’s accuracy and localization performance. Through the combination of these two techniques, the YOLOv5-SA-FC model achieved a better performance than the original YOLOv5 model. In particular, the proposed model detected objects with higher precision and recall while also being more efficient and robust with respect to variations in lighting, scale, and orientation.

## 4. Discussion

This paper introduces a more advanced version of the YOLOv5 model, called YOLOv5-SA-FC, which is specifically designed for the efficient detection of individual pigs. To demonstrate the effectiveness of our proposed model, we carried out a comparative study against several popular models, as well as ablating our model to obtain four different models: YOLOv5, YOLOv5-SA, YOLOv5-FC, and YOLOv5-SA-FC. Our experimental results indicated that both the YOLOv5-SA and YOLOv5-FC models outperform the original YOLOv5 model, thereby validating the effectiveness of both the Focal-CIoU and shuffle attention modules.

We also found some existing research used in the detection of pigs with YOLOv5 (e.g., Lai [38], Li [39], and Zhou [40]) and compared them with our method. The specific results are shown in Table 6 and were assessed using the mAP@0.5 metric:

From Table 6, it can be seen that our YOLOv5-SA performs the best. The shuffle attention uses dual-channel fusion technology and a channel shuffling operation, making the model more sensitive to capturing target features, which enables the model to adaptively extract valuable information in critical areas of the image, reduce irrelevant interference (such as overlapping pigs), and improve accuracy. Moreover, the YOLOv5-SA-FC model achieved the best performance in all comparisons, further demonstrating the superiority of our proposed model. By leveraging the shuffle attention module, our model could dynamically focus on the most relevant features for pig detection and counting while reducing the weights of non-essential features. Additionally, the Focal-CIoU loss mechanism gave a higher priority to predicted boxes having higher overlap with the target box, thus significantly improving the detection performance.

In addition, to translate our research findings into practical productivity, we require several essential practical steps. Firstly, it is necessary to ensure that the farm has a computing center with sufficient performance capabilities to efficiently handle the computational tasks required by the models. Secondly, real-time video feeds from each pigpen need to be transmitted to the computing center, which may involve laying a substantial amount of wiring on the farm or establishing a wireless network hub to ensure the timeliness of data received by the computing center. Finally, personnel training is essential for the farm to operate smoothly, maintain the system, and provide technical support when necessary. In summary, deploying the model into practical production requires consideration of numerous factors, such as software, hardware, personnel, etc., and their effective integration to ensure the system’s proper functioning.

In conclusion, our proposed YOLOv5-SA-FC model outperformed existing models in terms of accuracy and efficiency, making it a promising solution for pig detection and counting applications.

## 5. Conclusions

In this paper, we proposed a new pig detection and counting model called YOLOv5-SA-FC, which integrates shuffle attention and Focal-CIoU loss into the YOLOv5 framework backbone. The channel attention and spatial attention units in the shuffle attention module enable YOLOv5-SA-FC to effectively focus on the critical regions of the image with a high detection capability, thereby enhancing the feature extraction performance with more discriminative, robust, and rich feature maps. The Focal-CIoU loss mechanism forces the model to prioritize the predicted boxes having higher overlap with the target box, thereby increasing the contribution of positive samples and improving the detection performance. Furthermore, we conducted ablation studies, the results of which indicated the performance enhancements brought by both the shuffle attention and Focal-CIoU modules. Moreover, the experimental results indicated that the proposed YOLOv5-SA-FC model presents a promising pig detection and counting performance, with a 93.8% mAP and 95.6% accuracy, thus significantly outperforming other state-of-the-art models.

In future work, we plan to develop more sophisticated and advanced models to further enhance the pig detection and counting performance. We intend to explore various data augmentation methods to improve the robustness of the model and experiment with other attention mechanisms to capture crucial features for pig detection with better accuracy. Additionally, we hope to extend our work to more complex scenarios, including pig tracking and behavior analysis, in order to better understand their habits. Finally, we will investigate the potential of applying our proposed model to various animal detection and counting tasks aside from pig detection, thus expanding the scope of our research.

## Figures and Tables

**Figure 1 animals-13-03201-f001:**
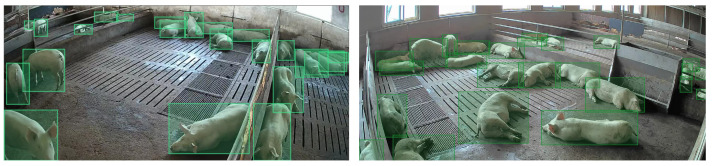
Some examples of annotated images.

**Figure 2 animals-13-03201-f002:**
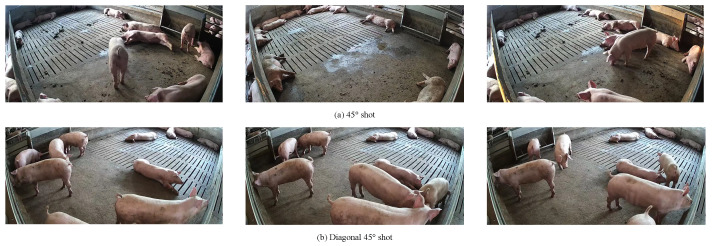
The collected data samples for pig detection and counting.

**Figure 3 animals-13-03201-f003:**
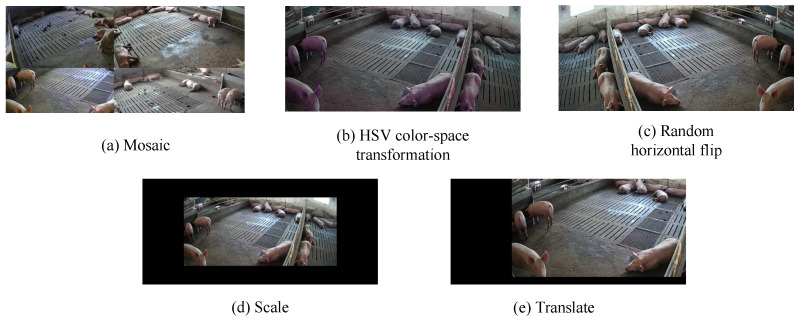
Some examples of data augmentation.

**Figure 4 animals-13-03201-f004:**
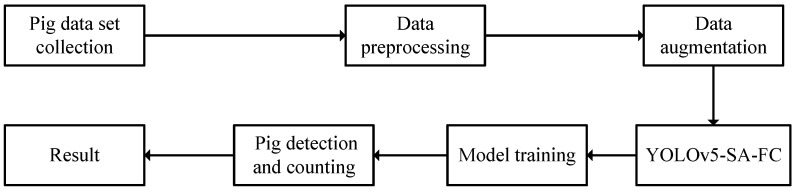
The technical route of YOLOv5-SA-FC for pig detection and counting.

**Figure 5 animals-13-03201-f005:**
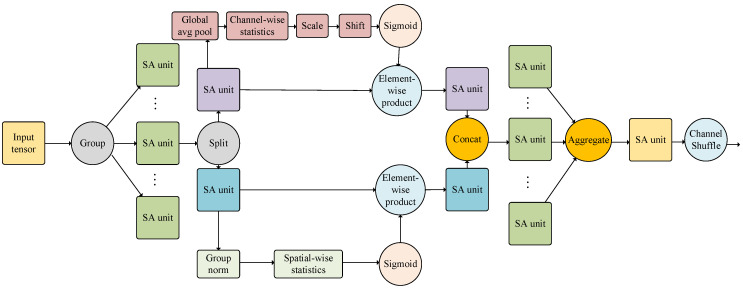
The structure of the shuffle attention module.

**Figure 6 animals-13-03201-f006:**
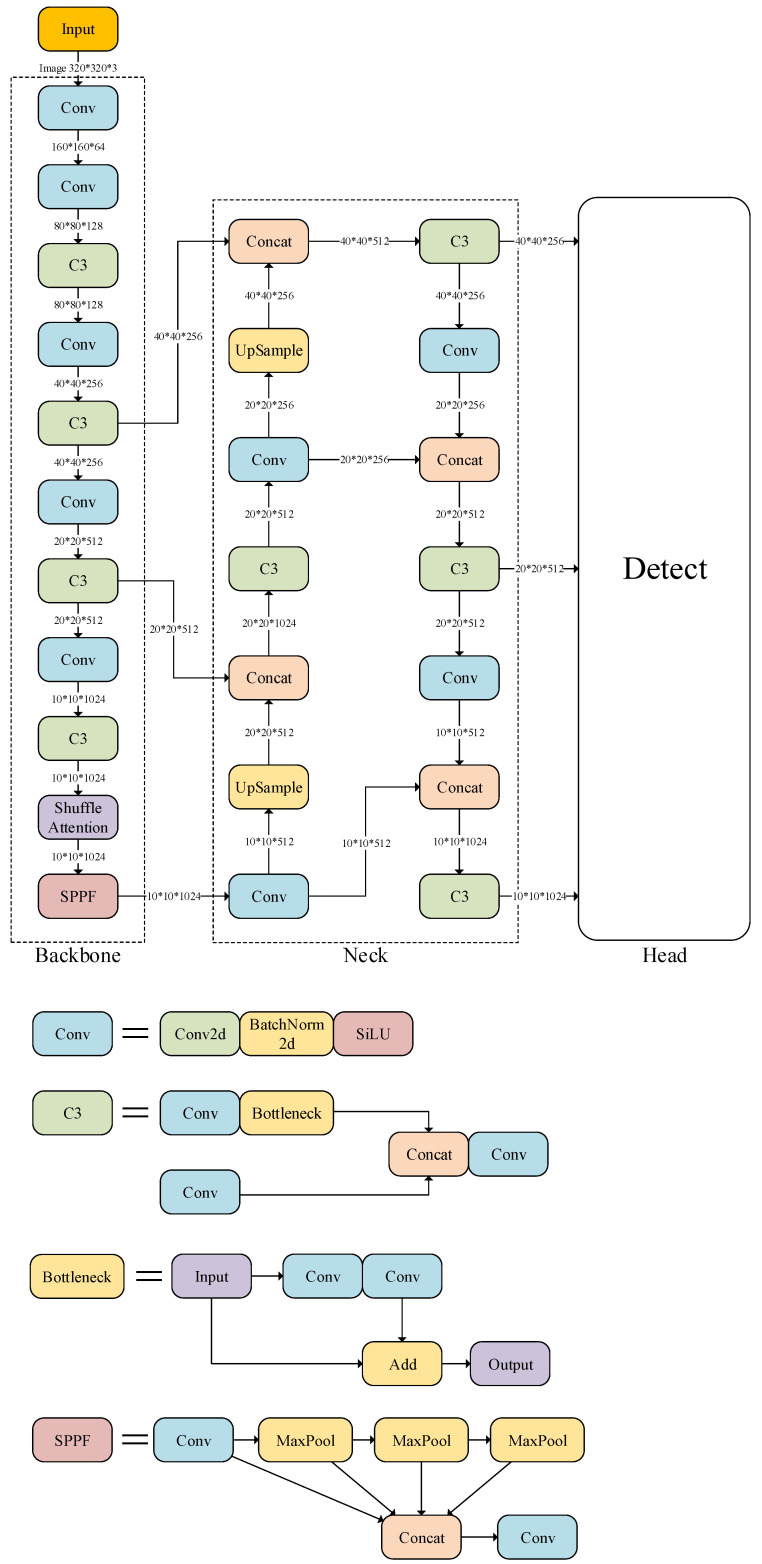
The pipeline of the proposed YOLOv5-SA-FC.

**Figure 7 animals-13-03201-f007:**
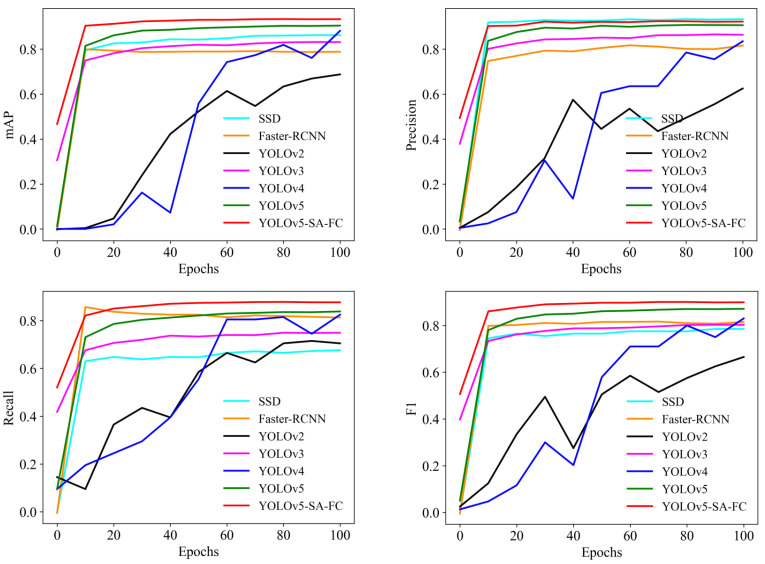
Comparison curves of different models, including the SSD, Faster-RCNN, YOLOv2, YOLOv3, YOLOv4, YOLOv5, and YOLOv5-SA-FC models, under different iteration times. The four subplots show the comparison curves of different models for mAP, precision, recall, and F1 score at different iterations.

**Figure 8 animals-13-03201-f008:**
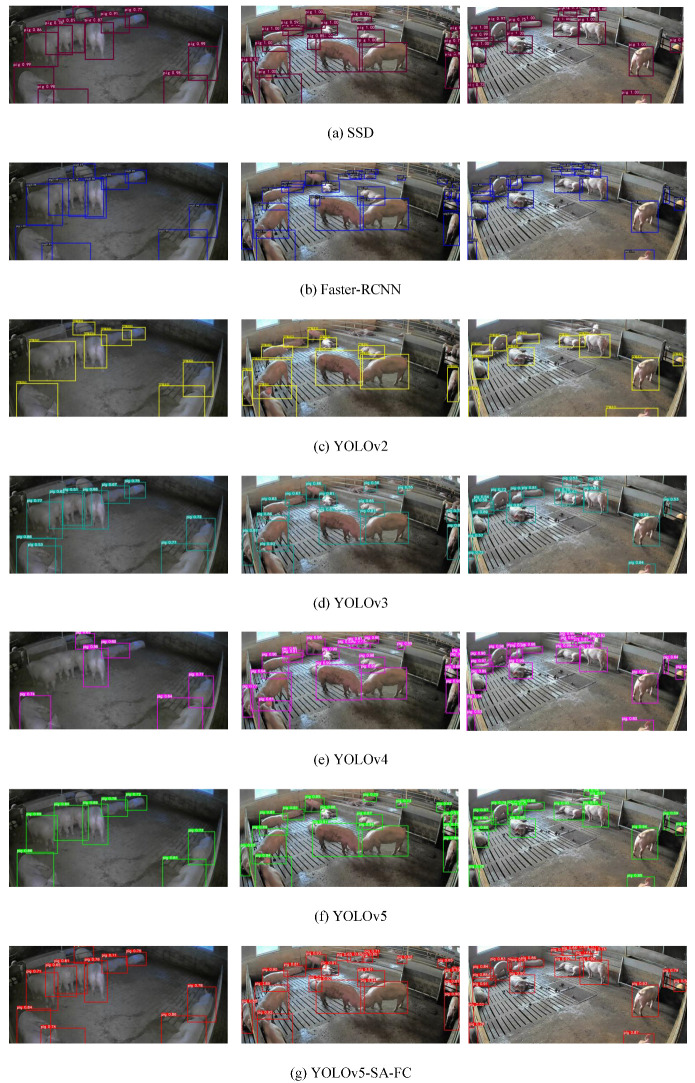
Qualitative comparison results of SSD, Faster-RCNN, YOLOv2, YOLOv3, YOLOv4, YOLOv5, and YOLOv5-SA-FC models.

**Figure 9 animals-13-03201-f009:**
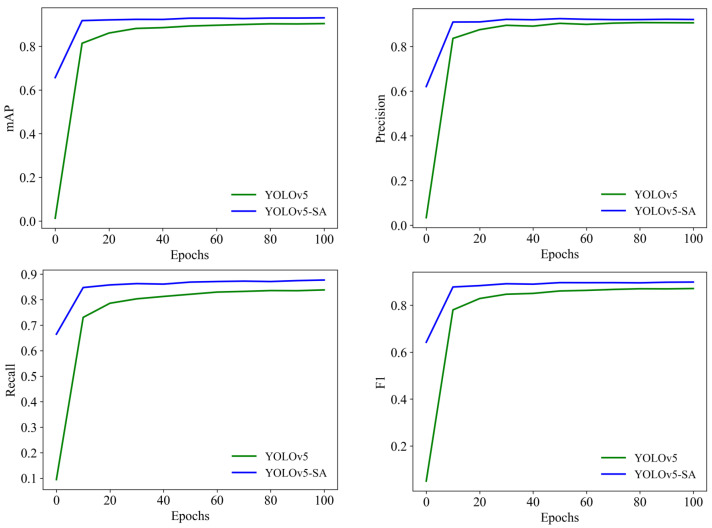
Comparison curves of different models, including YOLOv5 and YOLOv5-SA, under different iteration times. The four subplots show the comparison curves of different models for mAP, precision, recall, and F1 score at different iterations.

**Figure 10 animals-13-03201-f010:**
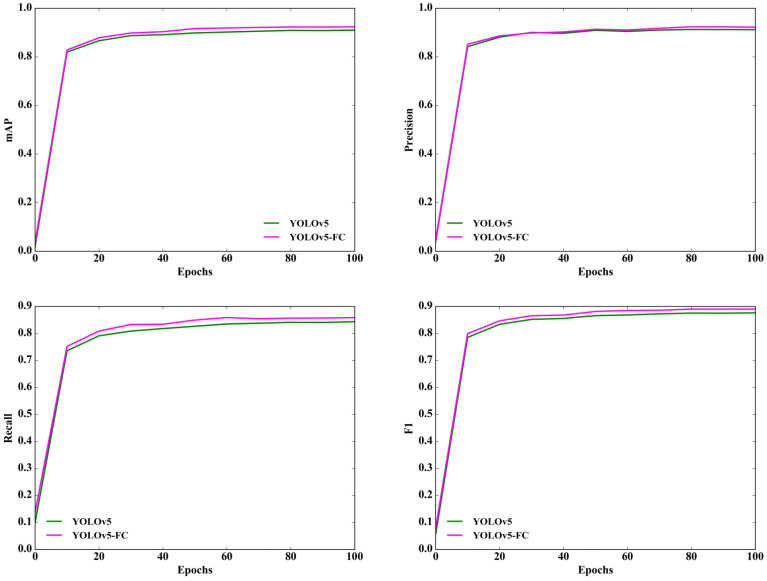
Comparison curves of different models, including YOLOv5 and YOLOv5-FC, under different iteration times. The four subplots show the comparison curves of different models for mAP, precision, recall, and F1 score at different iterations.

**Figure 11 animals-13-03201-f011:**
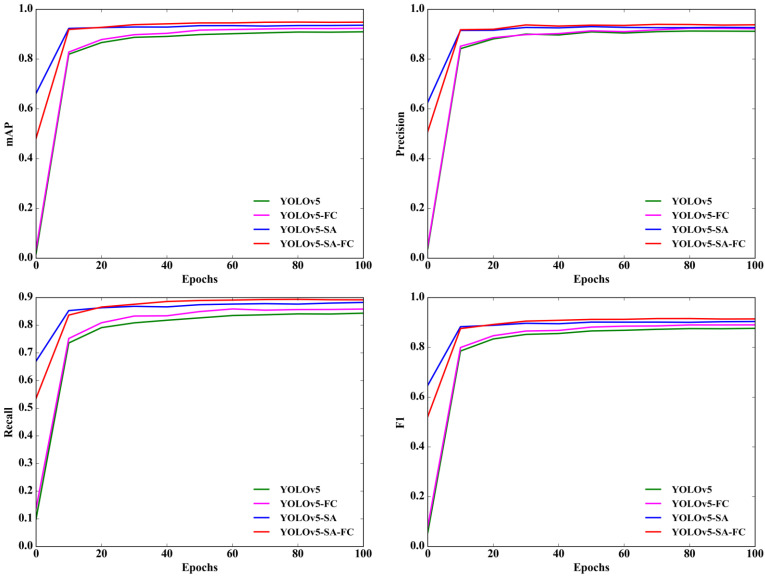
Comparison curves of different models, including YOLOv5, YOLOv5-FC, YOLOv5-SA, and YOLOv5-SA-FC, under different iteration times. The four subplots show the comparison curves of different models for mAP, precision, recall, and F1 score at different iterations.

**Figure 12 animals-13-03201-f012:**
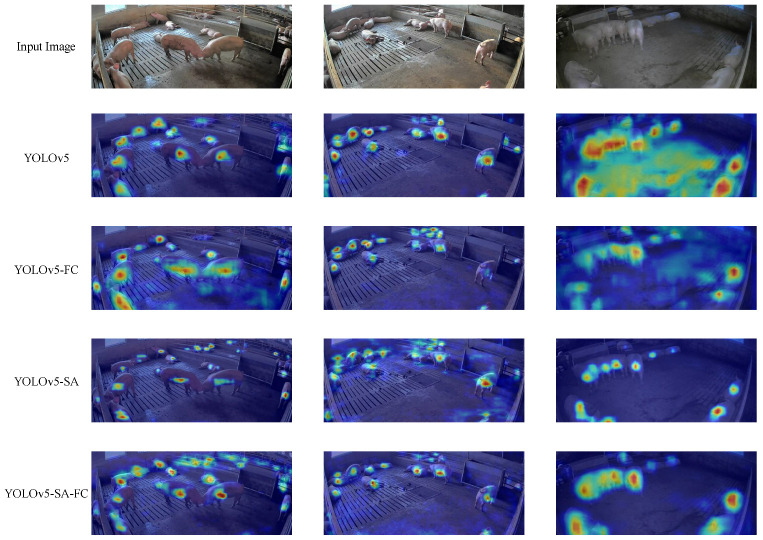
Comparison heatmaps of different models, including YOLOv5, YOLOv5-FC, YOLOv5-SA, and YOLOv5-SA-FC, respectively.

**Table 1 animals-13-03201-t001:** Configuration of hardware and software environment for experiments.

Term	Configurations
Operating System	Ubuntu 18.04
GPU	NVIDIA GeForce GTX TITANXP
CPU	Intel Core I7 7800 X
Memory	128GB
Hard disk	4TB SSD*3
Python	3.6.9
Pytorch	1.2.0
CUDA	11.2
CUDNN	10.0.130

**Table 2 animals-13-03201-t002:** Comparision of different models.

Model	Counting Accuracy	Precision (%)	Recall (%)	F1 Score	Mean Average Precision (%)
Faster-RCNN	-	82.10	81.80	0.81	89.5
SSD	-	93.8	68.1	0.79	86.6
YOLOv2	69.8	63	71	0.67	69.2
YOLOv3	79.8	86.9	75.4	0.80	83.6
YOLOv4	82.8	84.0	83.0	0.83	88.6
YOLOv5	86.2	91.0	84.3	0.87	91.0
YOLOv5-SA-FC	95.6	92.7	88.1	0.90	93.8

**Table 3 animals-13-03201-t003:** Comparision of YOLOv5 and YOLOv5-SA.

Model	Counting Accuracy	Precision (%)	Recall (%)	F1 Score	Mean Average Precision (%)
YOLOv5	86.2	91.0	84.3	0.87	91.0
YOLOv5-SA	94.4	92.3	87.9	0.89	93.6

**Table 4 animals-13-03201-t004:** Comparision of YOLOv5, YOLOv5-FC, YOLOv5-SA, and YOLOv5-SA-FC.

Model	Counting Accuracy	Precision (%)	Recall (%)	F1 Score	Mean Average Precision (%)
YOLOv5	86.2	91.0	84.3	0.87	91.0
YOLOv5-FC	88.8	92.2	85.8	0.89	92.3
YOLOv5-SA	94.4	92.3	87.9	0.89	93.6
YOLOv5-SA-FC	95.6	92.7	88.1	0.90	93.8

**Table 5 animals-13-03201-t005:** Comparision of YOLOv5 and YOLOv5-FC.

Model	Counting Accuracy	Precision (%)	Recall (%)	F1 Score	Mean Average Precision (%)
YOLOv5	86.2	91.0	84.3	0.87	91.0
YOLOv5-FC	88.8	92.2	85.8	0.89	92.3

**Table 6 animals-13-03201-t006:** Comparision of ECA, CA, CBAM, YOLOv5-SA, and YOLOv5-SA-FC.

	ECA [38]	CA [39]	CBAM [40]	YOLOv5-SA	YOLOv5-SA-FC
Mean Average Precision (%)	93.5	92.6	92.9	93.6	93.8

## Data Availability

The data sets generated during and/or analyzed during the current study are available from the corresponding authors on reasonable request.

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
