# Peer review of "YOLOv5-SA-FC: A Novel Pig Detection and Counting Method Based on Shuffle Attention and Focal Complete Intersection over Union"

_animals, 2023, doi:10.3390/ani13203201_

Round 1
Reviewer 1 Report
The model for pig detection and counting based on Shuffle Attention (SA) and Focal-CIoU enhanced YOLOv5 (YOLOv5-SA-FC) is proposed in this paper, but there are the following problems.
1、Please abbreviate section of 2.3.1, 2.3.2, 2.3.3, 2.3.4
2、Line179:6955 samples are used for training and 1115 samples are used for testing. What is the regulation for dividing the training and testing sets?
3、what is the characterstic for training data and testing data?
4、Please introduce the meanings of all variables and parameters in equation 14 to 18.
6、Line599,I consider that the discussions should be changed to discussion.
5、The discussion did not compare with other people's research,Please compare with the discussion of others‘ research.
Moderate editing of English language required
Reviewer 2 Report
The research proposed YOLOv5-SA-FC model to detect and counting pigs in a farm of Shanxi. The author would like to prove YOLOv5-SA-FC is an innovated method and outperformed similar researches. The data set collection and processing is sound. The introduction on the YOLOv5 and SA is clear. However, there is still some problems. 1) no single paragraph to introduce the FC model. 2) Why the author compared YOLOv5-SA-FC with Faster-RCNN, SSD, YOLOv2, YOLOv3, YOLOv4 and YOLOv5. It seems that it is obvious that YOLOv5-SA-FC outperformed other model since the author has modified it. The comparison is no needed. 3) The comparison between YOLOv5 and YOLOv5-SA or YOLOv5 and YOLOv5-FC still seems redundant. 4) I prefer to see the comparison of the YOLOv5 with the published research. How does your research outperformed them? 5) How to put the model into real production is still a long way. No such information presented in the manuscript.
In abstract, line 3, deep learning techniques don’t belongs to traditional methods.
Line 10, “93.8%/95.6%” should be “93.8% and 95.6%”.
In figure 5, some squares don’t have any words on it. It is hard to understand.
Lots of grammer issues. The author should check it carefully.
Reviewer 3 Report
Review comments of manuscript animals-2613006
The manuscript proposed an improved YOLOv5 model with shuffle attention and focal CIoU loss function to enhance the detection and counting of group-housed pigs.
- Give the appropriate citation for paragraphs 2 and 3 of the Introduction.
- The authors have mentioned that the pig barn floor is made of concrete, but in the sample images, it seems it is made of concrete and slatted materials. Please verify and correct it.
- Please mention the model of the cameras used to collect the dataset.
- The authors mentioned that they manually preprocessed the datasets, which is challenging for the real-time applications of the proposed system,
- Make Sense.ai online tool is used to label the dataset. Please explain how accurately the software labelled the datasets. Did you check through any means whether the images are correctly labelled or not?
- What is the logic behind splitting the images into training and testing images at 6955 and 1115? It is suggested to use a definite ratio (like 80:20 or 90:10).
- How effectively does the HSV transformation support data augmentation, as it is a new approach? If you have any reference that used other colour space transformations as data augmentation along with RGB image datasets, please mention it.
- Please ensure the YOLOv5 model has been cited and referred to correctly.
- How did the authors fix the SA position in the backbone network?
- Replaced ours by YOLOv5-SA-FC in the comparison table
- Replace P/%, R/%, mAP/% by only % in Tables 2-5.
- Please improve the contents written in the 3.4 section and discussion sections.
Please use standard presentation in result and discussion sections.
Round 2
Reviewer 1 Report
I consider this paper which manuscript ID is animals-2613006 can be published in present form.
Reviewer 2 Report
no more comments.
Moderate editing of English language required.
Reviewer 3 Report
Thank you for well addressing my comments.